# Pre-trained Neural Recommenders: Learning Statistical Representations for Zero-shot Recommender Systems

## Abstract

Modern neural collaborative filtering (NCF) techniques are critical to the success of e-commerce, social media, and content-sharing platforms. However, despite technical advances—for every new application domain, we need to train an NCF model from scratch. In contrast, pre-trained vision and language models are routinely applied to diverse applications directly (zero-shot) or with limited fine-tuning. Inspired by the impact of pre-trained models, we explore the possibility of pre-trained recommender models that support building recommender systems in new domains, with minimal or no retraining, without the use of any auxiliary user or item information. We propose a framework that computes the user and item representations via learning the representations of the user/item activity quantiles. With extensive experiments on five diverse datasets, we show that the framework can not only generalize to unseen users and unseen items within a dataset and across different datasets (*i.e.,* cross-domain, zero-shot) but with comparable performance to state-of-the-art neural recommenders.

## 1 Introduction

While neural recommenders achieve great success on online platforms, the training process of these neural models remains the same, requiring models to train from scratch for every new application domain. Further, even if we train the model for a domain, a large influx of new users and new items necessitates retraining for high performance. In contrast, in peer fields of AI, pre-trained vision models (Dosovitskiy et al., 2020; He et al., 2016; Radford et al., 2021), and pre-trained language models (Devlin et al., 2018; Brown et al., 2020; Raffel et al., 2020; Liu et al., 2019) have transformed how practitioners use these models. In some scenarios (Brown et al., 2020; Devlin et al., 2018; He et al., 2016; Raffel et al., 2020), a few fine-tuning steps are sufficient to achieve state-of-the-art performance (few-shot learning), while in other cases (OpenAI, 2023; Brown et al., 2020), practitioners use the pre-trained models directly without any fine-tuning (zero-shot learning). Thus in this paper, we ask: *Can we develop pre-trained recommender models (PRMs) with transferable representations, deployed to new domains, with minimal or no retraining?* We want these models to generalize to: (i) unseen users and unseen items within the same domain; (*e.g.,* recommendations when new users and items are added to a platform) (ii) new domains with no overlapping users, items, or auxiliary features with the training domain. (*e.g.,* using a model for movie recommendations in the United States for grocery recommendations in Japan.)

At first sight, developing a pre-trained recommender model with transferable representations appears impossible and conceptually distinct from creating a pre-trained language model. Language models learn transferable representations of a word conditioned on the past words within a text window to maximize the likelihood of the next word. Since the documents in the training corpus and the inference time applications share the same language(s), the models learn universal representations of a word and a conditional probability of occurrence given past words, which are then used to generate text or make inferences (Devlin et al., 2018; Brown et al., 2020). In contrast, recommender system datasets have two entities—users and items. Furthermore, unlike documents from the same language(s) sharing the same set of words, where direct word correspondences are possible, *we cannot form correspondences between users and items across different recommendation datasets, when the sets of users, items or both may be disjoint*. In this work, we show how to develop a

transferable basis for constructing user and item representations that accomplish the goals of the pre-trained recommender model solely based on the binary user-item interaction matrix without auxiliary data about the users or items. Next, we briefly discuss the related approaches.

Our target application goals are related to research on zero-shot, cold-start, and cross-domain recommendations. Zero-shot recommendation approaches typically rely on shared auxiliary user or item information (Ding et al., 2021; Feng et al., 2021; Sileo et al., 2022), *e.g.,* attributes and profiles. This information is typically unavailable or highly application-dependent, even when available. On the other hand, cross-domain recommender models are typically trained in information-rich source domains to improve the performance in information-sparse target domains (Man et al., 2017b; Hu et al., 2018; Li & Tuzhilin, 2020). This assumes the existence of overlapping users or items across domains (Cao et al., 2022; Li & Tuzhilin, 2020; Man et al., 2017a). Another approach is to exploit generic descriptions of users and items, *e.g.,* textual and visual descriptions, when tackling cross-domain recommendation with no user/item overlap (Liu et al., 2022; Kanagawa et al., 2019) and cold-start users or items (Lee et al., 2019; Dong et al., 2020). In this work, we develop a pre-trained recommender model for the zero-shot setting without relying on any shared auxiliary information.

**Present Work:** Our fundamental insight is that for developing successful zero-shot cross-domain recommender systems, learning transferable representations of the *statistical characteristics* of users and items, from the binary interaction data (which are available for all domains), is critical. Prior work indicates that the marginals of user and item activity (Barabasi, 2005; Salganik et al., 2006) follow heavy-tailed distributions and that the nature of these distributions arises due to human activity. We observe qualitatively similar heavy-tailed distributions across recommender system domains (Krishnan et al., 2018; Abdollahpouri et al., 2020). In this work, we learn the representations of each activity quantile, where the quantile could represent user or item activity. We regularize the learned representations by enforcing quantile ordering and smoothness constraints. In our pre-trained recommender model (QRec), we map these learned quantile representations into user and item embeddings by using the target interaction matrix. We show excellent zero-shot performance without auxiliary data, on real-world datasets. We use regular in-domain recommendation tasks (not our focus) to understand the utility of quantile representations. Impressively, we observe that QRec can account for 85%–93% (AUC) of the state-of-the-art performance (He et al., 2020) for regular in-domain recommendations (table 2). This suggests that our quantile representations capture most of the information in the interaction matrix. For both in-domain and cross-domain zero-shot recommendations, the focus of this paper, (impossible for state-of-the-art methods without auxiliary data), QRec's excellent performance matches its in-domain performance (table 3). QRec is stable while neural model performance degrades significantly with smaller training size (fig. 1). Furthermore, QRec is small sized ($O(d^2)$ vs. $O(d \times (\mathbb{U} + \mathbb{V}) + d^2)$ for neural models, with $\mathbb{U}, \mathbb{V} \gg d$) (appendix A.5) and trains 1.5–14× faster than neural models (Appendix, table 5). Our contributions:

*Learning activity quantile representations:* We are the first to propose learning transferable representations of the activity quartiles, for zero-shot recommender systems without auxiliary information. We learn representations of the activity quantiles, and regularize them by enforcing constraints due to quantile ordering and representational smoothness. These quantile representations are crucial to zero-shot knowledge transfer without auxiliary information.

*Pre-trained recommender models:* We are the first to propose pre-trained recommender systems for in-domain and cross-domain zero-shot scenarios. In contrast, standard neural recommenders (Rendle et al., 2009; He et al., 2017; Wang et al., 2019b; He et al., 2020) must be trained on each new dataset, and learn non-transferable representations for each user and item. Our model derived representations of users and items by using the interaction matrix to map representations of the activity quantiles. A pre-trained model will have a significant application impact as we can use it to develop a recommender system for a new application domain with minimal fine-tuning (few-shot) or no training (zero-shot).

*Zero-zhot without auxiliary data:* We propose a novel, challenging setting for recommender systems: zero-shot recommendations on unseen users and items within and across domains without auxiliary information. In contrast, prior work on zero-shot recommendation or cold-start (Lee et al., 2019; Dong et al., 2020; Ding et al., 2021; Feng et al., 2021; Yuan et al., 2020) assumes the presence of additional auxiliary information while cross-domain transfer (Man et al., 2017a; Zhao et al., 2020; Zhu et al., 2021) assumes overlapping users or items. To the best of our knowledge, we are the first to tackle this challenging knowledge-transfer setting in recommender systems.

## 2 RELATED WORK

In this section, we briefly review a few related lines of work that is relevant to this paper.

**Neural Collaborative Filtering**: The core idea of latent-factor collaborative filtering (CF) methods is to learn latent representations of users and items to maximize the likelihood of observed user-item interactions. Neural CF (NCF) methods enhance the capability of traditional CF methods with non-linear latent factor interactions—*e.g.,* via interaction modeling (Rendle et al., 2009; He et al., 2017; Tay et al., 2018), or graph convolution on the interaction graph (Wang et al., 2019b; He et al., 2020). However, their recommendation results rely heavily on historical interactions between users and items. While some NCF methods handle unseen items or unseen users (Liang et al., 2018; Sankar et al., 2021) when they interact with the existing users or items, they cannot use historical interactions to make predictions when *both* the users and items are unseen. In contrast, our pre-trained recommender framework can generalize to unseen users and items within the same domain or to new domains without model retraining.

**Cross-domain Recommendation**: The general goal of cross-domain recommendation is to leverage relatively data-rich domains to improve the recommendation performance of the data-sparse domain(Hu et al., 2018; Li & Tuzhilin, 2020; Man et al., 2017b). However, most of this work assumes that there are overlapping users and items (Man et al., 2017a; Zhao et al., 2020; Zhu et al., 2021) for effective knowledge transfer and domain adaptation.

**Cold-start Recommendation**: Methods have been proposed to tackle either the cold-start item case (Volkovs et al., 2017; Wei et al., 2021; Du et al., 2020; Liu et al., 2021; Huan et al., 2022) or the cold-start user case (Lu et al., 2020; Zhu et al., 2021; Feng et al., 2021; Yuan et al., 2020). However, they rely on auxiliary information to generate the initial representation and can handle only one-sided (*i.e.,* either user or item) cold-start problems. Some past work explores meta-learning for in-domain cold-start (Pang et al., 2022; Hao et al., 2021) and cross-domain recommendation tasks on unseen users and unseen items (Lee et al., 2019; Dong et al., 2020). However, they either rely on application-dependent side information or require an additional fine-tuning step to learn the representations. The primary goal of this paper is *not* to address the cold-start recommendation problem. Instead, we develop a highly generalizable, pre-trained recommender framework that can zero-shot transfer to any new domain with just the user-item interaction matrix and without any auxiliary information.

**Zero-shot Learning**: Zero-shot learning originated in computer vision to make predictions on previously unseen test classes (Wang et al., 2019a; Feng et al., 2021). A general approach is to make image predictions based on the images' semantic information. Analogously, some zero-shot approaches on recommender systems try to map the attribute space of users and items into their latent space (Ding et al., 2021; Feng et al., 2021; Lee et al., 2019; Dong et al., 2020). However, they all rely on highly dataset-dependent features. In contrast, our work relies only on the binary user-item interaction matrix, present in all recommendation datasets.

To summarize, prior work in recommender systems has focused on learning representations for *each* user and item in the domain. To predict interactions for unseen users or items, they assume overlap in either users, or items with training data, or if there is no overlap, the presence of auxiliary information. In contrast, we learn representations of activity quantiles, and develop a pre-trained recommender framework that can generalize to unseen users *and* items (*i.e.,* zero-shot) within or across different domains without any auxiliary information (*e.g.,* user or item descriptions).

## 3 PROBLEM DEFINITION

In this paper, we consider our recommendation problems with only implicit feedback, *i.e.,* no explicit ratings or auxiliary information. We denote the user set as $\mathbb{U}$, the item set as $\mathbb{V}$, and the binary interaction matrix as $\boldsymbol{M} \in \mathbb{R}^{|\mathbb{U}| \times |\mathbb{V}|}$. Neural collaborative filtering methods essentially train a recommender system $\mathcal{R}$ that learns the latent representations for every user $u \in \mathbb{U}$ and item $v \in \mathbb{V}$. The overall goal of training $\mathcal{R}$ in the traditional recommender systems is to learn a personalized scoring function $\mathcal{F}(v|u, \boldsymbol{M})$ that captures the likelihood of $v$ interacting with $u$.

In this paper, we focus on the generalizability of the statistical representations. Therefore, we consider the following two recommendation problem formulations:

| Symbol | Description |
|---|---|
| $P(a_u), P(a_v)$ | probability of interaction for users and items |
| $r_{u_i}, r_{v_j}$ | activity percentile of user $u_i$ and item $v_j$ |
| $\mathcal{N}_{u_i}, \mathcal{N}_{v_j}$ | the interacted percentiles of user $u_i$ and item $v_j$ |
| $d$ | dimension of the percentile representation |
| $Q$ | learnable look-up matrix for activity percentile representation |
| $T$ | positional embedding |
| $W_u, W_v$ | learnable weight matrices |
| $h$ | learnable weight matrix for the attention function |
| $e_{u_i}, e_{v_j}$ | representation of user $u_i$ and item $v_j$, derived from, $Q, M, W_u, W_v$ and $h$ |
| $\lambda$ | hyper-parameter for the weighting regularization loss |

Table 1: Notation

Let $\mathbb{U}$ and $\mathbb{V}$ be the user and item set of an interaction matrix $M$ that neural collaborative filtering methods are trained on.

**Problem 1 (Zero-shot, In-domain Recommendation)** *Given an interaction matrix $M'$ over $\mathbb{U}'$ and $\mathbb{V}'$, where $M'$ and $M$ belong to the same domain, $\mathbb{U}' \cap \mathbb{U} = \varnothing$, and $\mathbb{V}' \cap \mathbb{V} = \varnothing$, learn a recommender system $\mathcal{R}'$ that produce a scoring function $\mathcal{F}'(v'|u', M')$ for $v' \in \mathbb{V}'$ and $u' \in \mathbb{U}'$ without training the scoring function on $M'$.*

**Problem 2 (Zero-shot, Cross-domain Recommendation)** *Given an interaction matrix $M^*$ over $\mathbb{U}^*$ and $\mathbb{V}^*$, where $M^*$ and $M$ **belong to different domains**, $\mathbb{U}^* \cap \mathbb{U} = \varnothing$, and $\mathbb{V}^* \cap \mathbb{V} = \varnothing$, learn a recommender system $\mathcal{R}^*$ that produce a scoring function $\mathcal{F}^*(v^*|u^*, M^*)$ for $v^* \in \mathbb{V}^*$ and $u^* \in \mathbb{U}^*$ without training the scoring function on $M^*$ directly.*

Addressing problems 1 and 2, zero-shot in-domain and cross-domain scenarios, without auxiliary data, requires us to learn novel, transferable representations and to develop a framework that can incorporate such representations. We discuss our approach in the next section.

## 4   PRE-TRAINED RECOMMENDER SYSTEM FRAMEWORK

In this section, we show how to use the representations of activity percentiles for deriving user and item representations in § 4.1, the training regimen in § 4.2 and zero-shot model inference in § 4.3.

### 4.1   LEARNING STATISTICAL REPRESENTATIONS OF USERS AND ITEMS

Motivated by prior work in network science (Barabási & Albert, 1999; Barabasi, 2005), recent work in recommender systems (Krishnan et al., 2018), and experimental work (Salganik et al., 2006), that suggests ubiquity of heavy-tailed distributions, we propose to represent users (items) using learned representations of the activity percentile of the items (users) they interact with.

Formally, given an interaction matrix $M$, first, we determine marginal user $P(a_u)$ and item interaction probability $P(a_v)$ distributions, where for example, $P(a_{u_i})$ is proportional to the count of the interactions of user $u_i$. Then, using $P(a_u)$ and $P(a_v)$, we can determine for any user $u_i$ or item $v_j$ its activity percentile $r_{u_i}$ and $r_{v_j}$, where $r \in \{0, 1, ..., 100\}$. Thus, we want to learn the representation of each activity percentile $r$ (independent of users and items) but use them to compute user and item representations via a mapping function. Let $Q \in \mathbb{R}^{100 \times d}$ be a learnable look-up matrix and denote $q_r = Q_{r,:}$ as the representation of percentile $r$. To impose order on the learned representations of percentiles, we use positional encoding $T \in \mathbb{R}^{100 \times d}$ (see appendix A.1), specific to the percentile $r$ (*i.e.*, $t_r = T_{r,:}$ is the r-th row of $T$). We compute the representation of user $u_i$ as:

$$e_{u_i} = \sum_{r \in \mathcal{N}_{u_i}} \alpha_{u_i,r}(m_{u_i,r}q_r + t_r), \quad \alpha_{u_i,r} = \frac{\exp(h^T(W_v(m_{u_i,r}q_r + t_r)))}{\sum_{r' \in \mathcal{N}_{u_i}} \exp(h^T(W_v(m_{u_i,r'}q_{r'} + t_{r'})))} \quad (1)$$

where $h \in \mathbb{R}^d$ is the attention function, $W_v \in \mathbb{R}^{d \times d}$ is a learnable weight matrix. Note that $t_r$ is the pre-computed positional encoding of the percentile $r$ and $m_{ij}$ is a pre-computed constant that

indicates the number of times user $u_i$ has interacted with items in the percentile $r_{v_j}$. What the first term of eq. (1) says is that the user embedding is determined by the summing over the representations of the percentile $r$ (*i.e.*, $\boldsymbol{q}_r$) of the items that the user has interacted with, weighted by the attention $\alpha_{u_i,r}$ that the user pays to items in percentile $r$. The attention is computed as the softmax of the dot product between $\boldsymbol{h}$ and the projection of $(m_{u_i,r}\boldsymbol{q}_r + \boldsymbol{t}_r)$ on the weight matrix $\boldsymbol{W}_v$. Similarly, we compute the representation of item $v_j$ as:

$$\boldsymbol{e}_{v_j} = \sum_{r \in \mathcal{N}_{v_j}} \beta_{v_j,r}(m_{v_j,r}\boldsymbol{q}_r + \boldsymbol{t}_r), \quad \beta_{v_j,r} = \frac{\exp(\boldsymbol{h}^T(\boldsymbol{W}_u(m_{v_j,r}\boldsymbol{q}_r + \boldsymbol{t}_r)))}{\sum_{r' \in \mathcal{N}_{v_j}} \exp(\boldsymbol{h}^T(\boldsymbol{W}_u(m_{v_j,r'}\boldsymbol{q}_{r'} + \boldsymbol{t}_{r'})))} \tag{2}$$

similar to $\boldsymbol{W}_v$, $\boldsymbol{W}_u \in \mathbb{R}^{d \times d}$ is also a learnable weight matrix. Unlike traditional recommender models, and critical to support zero-shot recommendations, we do not learn representations for each user or item based on their IDs. Instead, we learn the representations of the percentiles $Q$ of the activity distribution of users and items. We derive the user and item representations from the percentiles via a mapping function.

## 4.2 MODEL TRAINING

Now we show how to use the activity percentile representations derived from the interaction matrix to develop a pre-trained model that can be deployed in a zero-shot scenario. We begin with model training, then we show how to use the pre-trained model for inference.

Given the statistical representation of user $u_i$ and $v_j$, computed with eqs. (1) and (2), we can compute the prediction of the interaction $\hat{y}(u_i \mid v_j)$ between user $u_i$ and item $v_j$ as the inner product of their representations:

$$\hat{y}(u_i \mid v_j) = P(a_{v_j}) \langle \boldsymbol{e}_{u_i}, \boldsymbol{e}_{v_j} \rangle \tag{3}$$

where $P(a_{v_j})$ is the probability that item $v_j$ will be in an interaction, which is proportional to the count of the number of interactions of item $v_j$. Since the activity distribution is heavy-tailed, normalizing the interaction count is necessary to avoid biasing the prediction towards popular users and items. We use the sigmoid function to normalize the interaction count. Thus, we operationalize $P(a_{v_j}) = \sigma(c(v_j) - \text{med}(c_v))$, where $c(v_j)$ is the number of interactions of item $v_j$, $\text{med}(c_v)$ is the median of the interaction count of all the items and where $\sigma$ is the sigmoid function. Notice that the scoring function $\mathcal{F}(v \mid u, \boldsymbol{M}) = \hat{y}(u_i \mid v_j)$, when the model is trained on domain $\boldsymbol{M}$.

We employ the *Bayesian Personalized Ranking*(BPR) loss proposed by Rendle et al. (2009) as our objective function to train the model parameters, where the loss $L_{BPR}$ defined as follows:

$$L_{BPR} = -\sum_{u \in \mathbb{U}} \sum_{v^+ \in \mathcal{G}_u} \sum_{v^- \notin \mathcal{G}_u} \ln \sigma(\hat{y}(u \mid v^+) - \hat{y}(u \mid v^-)) \tag{4}$$

where, $v^+$ and $v^-$ are positive and negative samples respectively, $\mathcal{G}_u$ is the set of items that user $u$ has interacted with.

Intuitively, we should expect that the representations of adjacent percentiles are more similar than representations for percentiles that are greatly separated. Thus we also impose a regularization on $Q$ to smooth the representation of the activity percentile $r$, by seeking to minimize $||\frac{dq}{dr}||$. Since $r$ is discrete, we approximate the derivative using an average of the forward difference. Formally, we define the regularization loss $L_{reg}$ as:

$$L_{reg} = \sum_{r} ||q_{r+1} - q_r|| \tag{5}$$

$$L = L_{BPR} + \lambda L_{reg}, \tag{6}$$

where, $L$ is the overall loss function, $\lambda$ is a tunable parameter. We use Adam (Kingma & Ba, 2014) as the optimizer in a mini-batch manner.

### 4.3 INFERENCE WITH THE PRE-TRAINED MODEL

Our inference setup for both zero-shot problems defined in § 3 is to compute the user/item activity percentile on the unseen interaction matrix and directly apply the model pre-trained on the seen interaction matrix to get the prediction.

Formally, suppose we are given a pre-trained model $\mathcal{F} : \mathbb{R} \times \mathbb{R} \to \mathbb{R}$ trained on $\boldsymbol{M}$, which includes all the parameters that are optimized in eq. (6), (*i.e.,* $\boldsymbol{Q}, \boldsymbol{W}_v, \boldsymbol{h}$). $\mathcal{F}(v \mid u, \boldsymbol{M})$ is the scoring function (formalized in § 3), learned from domain $\boldsymbol{M}$. Let us denote the interaction matrix of unseen users and unseen items as $\boldsymbol{M}'$. First, for each user and item in domain $\boldsymbol{M}'$, we compute the activity percentile $r$ for each user and item. Then we compute the prediction of the interaction $\hat{y}(v' \mid u')$ for item $v'$ given $u'$ as:

$$\hat{y}(v' \mid u') = \mathcal{F}(v' \mid u', \boldsymbol{M}') \tag{7}$$

Note that in this procedure, we use the pre-trained model $\mathcal{F}$, trained on domain $\boldsymbol{M}$ used for inference in domain $\boldsymbol{M}'$. The activity percentile $r$ computed from the interaction matrix $\boldsymbol{M}'$ of the unseen users and unseen items. We do not use any auxiliary information of the users and items. In addition, none of the parameters in $\mathcal{F}$ are updated during the inference.

To summarize, in this section, we showed how to develop a pre-trained recommender model based on learning the representations of the activity percentiles. We enforced ordinal structure on the representations and we regularized them to ensure smoothness. Model parameters learned in training are used to predict interactions in the zero-shot recommendation scenario with minimal knowledge of the target domain (*i.e.,* only the target interaction matrix) and without auxiliary information. We will evaluate the proposed model in the next section.

## 5 EXPERIMENTS

We present extensive experiments on five real-world datasets to evaluate the effectiveness of the pre-trained recommendation framework (named QRec ) under two problem settings introduced in § 3. We introduce the following research questions (RQ) to guide our experiments. RQ1: How well can QRec perform with regular, in-domain recommendation task setting against state-of-the-art neural recommenders? This question will help us quantify the degree to which activity representations can account for recommender system performance. RQ2: Can the learned statistical representation $\boldsymbol{Q}$ generalize to the zero-shot in-domain and cross-domain recommendation scenario? RQ3: What is the effect of the amount of training data available on QRec's in-domain and zero-shot performance?

### 5.1 DATASETS

We conducted experiments on five publicly available benchmark datasets from different domains and applications, *i.e.,* Epinions[1], Yelp[2], MovieLens 1M[3], Amazon-sport, and Amazon-Cloth (Ni et al., 2019). Detailed descriptions are in appendix A.2.

For the preprocessing step, first, we randomly partition the user and item set from each dataset into two equally sized parts, *i.e.,* seen users and unseen users, seen items and unseen items. Then, we select the corresponding seen users-seen items and unseen users-unseen items to construct the dataset for regular in-domain recommendation and zero-shot in-domain recommendation respectively. We preprocess each partitioned dataset under the 5-core setting (He & McAuley, 2016), to retain users and items with at least five interactions. In addition, for each user, we select 70% of its interaction to construct the training set, and the remaining 30% constitute the test set. From the training set, we randomly select 10% of the interactions as the validation set for hyper-parameter tuning and construct the proposed features for each user and item based on the remaining interaction in the training set. We use the training interactions to compute dataset statistics. The reason why we split the data by percentage is to maximally preserve the original data distribution. The statistics of the datasets after pre-processing are summarized in table 4.

---

[1]https://www.cse.msu.edu/tangjili/datasetcode/truststudy.html
[2]https://www.yelp.com/dataset
[3]https://grouplens.org/datasets/movielens

| DATASET | EPINIONS | | | YELP | | | MOVIELENS | | | CLOTH | | | SPORT | | |
| Metric | AUC | R@10 | N@10 | AUC | R@10 | N@10 | AUC | R@10 | N@10 | AUC | R@10 | N@10 | AUC | R@10 | N@10 |
|---|---|---|---|---|---|---|---|---|---|---|---|---|---|---|---|
| REGULAR IN-DOMAIN RECOMMENDATION | | | | | | | | | | | | | | | |
| BPR | 0.739 | 0.416 | 0.247 | 0.787 | 0.470 | 0.268 | 0.897 | 0.699 | 0.442 | 0.737 | 0.451 | 0.356 | 0.657 | 0.316 | 0.198 |
| NCF | 0.742 | 0.425 | 0.250 | 0.793 | 0.479 | 0.275 | 0.896 | 0.701 | 0.445 | 0.732 | 0.450 | 0.357 | 0.672 | 0.335 | 0.206 |
| NGCF | 0.761 | 0.462 | 0.269 | 0.812 | 0.513 | 0.298 | 0.901 | 0.706 | 0.451 | 0.758 | 0.496 | 0.389 | 0.714 | 0.386 | 0.245 |
| LightGCN | 0.808 | 0.526 | 0.324 | 0.824 | 0.560 | 0.330 | 0.903 | 0.712 | 0.456 | 0.767 | 0.524 | 0.410 | 0.765 | 0.467 | 0.303 |
| MostPop | —— | 0.358 | 0.208 | —— | 0.386 | 0.215 | —— | 0.527 | 0.312 | —— | 0.483 | 0.329 | —— | 0.366 | 0.216 |
| **QRec** | 0.706 | 0.381 | 0.214 | 0.741 | 0.394 | 0.218 | 0.839 | 0.531 | 0.313 | 0.772 | 0.490 | 0.332 | 0.694 | 0.377 | 0.218 |
| ZERO-SHOT IN-DOMAIN RECOMMENDATION | | | | | | | | | | | | | | | |
| MostPop | —— | 0.371 | 0.217 | —— | 0.420 | 0.238 | —— | 0.510 | 0.294 | —— | 0.487 | 0.332 | —— | 0.370 | 0.213 |
| **QRec** | 0.712 | 0.389 | 0.220 | 0.753 | 0.428 | 0.242 | 0.833 | 0.513 | 0.296 | 0.772 | 0.493 | 0.336 | 0.694 | 0.379 | 0.214 |

Table 2: QRec performance for the *regular in-domain* and *zero-shot in-domain* recommendation settings, introduced in § 3. While the regular in-domain recommendation is not the focus of the paper, we present the results to understand the degree to which QRec with quantile representations can account for the performance of state-of-the-art models. Impressively, QRec with a fraction of the training time (table 5), can retain on average, 92% of the AUC performance compared to the best-performing baselines in the regular in-domain recommendation setup. QRec also shows great generalizability in the zero-shot in-domain recommendation setting, outperforming the popularity-based baseline by up to 5%. QRec's performance gain over MostPop is higher for the most active users as seen in the appendix, fig. 2. Note, for zero-shot recommendations, we train on the seen user-seen item subset of the domain and make predictions using eq. (7) on the unseen user-unseen item subset of the same domain.

## 5.2 BASELINES & EXPERIMENTAL SETUP

We adopt four state-of-the-art collaborative filtering methods for benchmarking the performance of QRec, *i.e.,* BPR (Rendle et al., 2009), NCF (He et al., 2017), NGCF (Wang et al., 2019b), and LightGCN (He et al., 2020). In addition, to the best of our knowledge, we are the first to tackle the zero-shot recommendation setting with no side information. Therefore, we adopt an item popularity-based method (MostPop (Rendle et al., 2009; He et al., 2017; Kanagawa et al., 2019)) to produce the rank list by ranking the items based on their interaction counts. Detailed descriptions of the baselines are in appendix A.3.

We evaluate the effectiveness of our proposed features under the recommendation problems introduced in § 3 : 1) zero-shot in-domain recommendation; 2) zero-shot cross-domain recommendation. In addition, we also evaluate the performance of QRec under the regular recommendation problem setting to compare against selected state-of-the-art baselines.

We adopt a similar evaluation method to Rendle et al. (2009); He et al. (2017); Li et al. (2020). For each test interaction (user-item pair), we sample 99 unobserved items according to the user's interaction history. Then, we rank the test item for the user from the interaction among the 100 total items. We use AUC, NDCG@10, and Recall@10 as evaluation metrics on the produced rank lists. We compute the AUC score of the ranked list by giving unobserved interactions negative labels and actual test interactions positive labels. A higher AUC score indicates better ranking quality. Recall@10 measures if the given test interaction is within the top 10 out of the 100 items, and NDCG accounts for the ranking position of the test interaction. Random guessing will have an AUC score of 0.5 and a Recall@10 of 0.1. We calculate all three metrics and average the score of every test user. We report the average of five experimental runs.

All experiments were conducted on a Tesla V100 using PyTorch. We pre-train the recommender model for a maximum of 25 epochs using Adam optimizer. Similar to (He et al., 2020; Wang et al., 2019b), we fixed the embedding size to 64 for all the baselines. We train the LightGCN using two GCN layers and tune baselines in the hyper-parameter ranges centered at the author-provided values.

## 5.3 EXPERIMENTAL RESULTS

*In-domain recommendation (**RQ1**):* We investigate the performance of QRec against state-of-the-art methods under the regular in-domain and zero-shot in-domain recommendation setting (Table 2). First, let's examine the regular, in-domain recommendation results table 2. We emphasize that *with QRec, we aim not to outperform state-of-the-art NCF models within a domain.* Instead, our goal is to characterize the degree to which activity-based representations can explain the state-of-

| S → T Metric | EPINIONS AUC | R@10 | N@10 | YELP AUC | R@10 | N@10 | MOVIELENS AUC | R@10 | N@10 | CLOTH AUC | R@10 | N@10 | SPORT AUC | R@10 | N@10 |
|---|---|---|---|---|---|---|---|---|---|---|---|---|---|---|---|
| | REFERENCE: REGULAR IN-DOMAIN RECOMMENDATION | | | | | | | | | | | | | | |
| MostPop | —— | 0.358 | 0.208 | —— | 0.386 | 0.215 | —— | 0.527 | 0.312 | —— | 0.483 | 0.329 | —— | 0.366 | 0.216 |
| QRec | 0.706 | 0.381 | 0.214 | 0.741 | 0.394 | 0.218 | 0.839 | 0.531 | 0.313 | 0.772 | 0.490 | 0.332 | 0.694 | 0.377 | 0.218 |
| | CROSS-DOMAIN RECOMMENDATION, SOURCE →TARGET | | | | | | | | | | | | | | |
| Epinions'→ | —— | —— | —— | **0.739** | **0.392** | **0.217** | **0.835** | 0.528 | 0.314 | 0.730 | 0.479 | 0.325 | 0.691 | 0.368 | 0.217 |
| Yelp'→ | 0.687 | 0.372 | 0.212 | —— | —— | —— | 0.834 | **0.529** | **0.315** | 0.760 | 0.487 | 0.330 | 0.697 | 0.374 | 0.217 |
| MovieLens'→ | 0.659 | 0.353 | 0.196 | 0.711 | 0.377 | 0.205 | —— | —— | —— | 0.729 | 0.482 | 0.326 | 0.678 | 0.360 | 0.208 |
| Cloth'→ | 0.690 | 0.366 | 0.204 | 0.738 | 0.389 | 0.206 | 0.823 | 0.509 | 0.295 | —— | —— | —— | **0.698** | **0.375** | **0.217** |
| Sport'→ | **0.692** | **0.373** | **0.212** | 0.738 | 0.390 | 0.216 | 0.833 | 0.523 | 0.307 | **0.767** | **0.487** | **0.331** | —— | —— | —— |

Table 3: *Cross-domain, zero-shot recommendation:* The purpose of this experiment is to see if QRec can generalize across datasets with minimal performance loss. The notation $S \rightarrow T$ implies that we pre-train the recommender system using S's data (first column rows) and directly test on T (result in the corresponding row). To ensure a fair comparison, we train QRec using the seen user-seen-item training subset of the source domain $M$ and test on $M'$ using eq. (7). The test set is the same as the set used by baselines for in-domain recommendations for that domain. We make bold the best-performing cross-domain zero-shot transfer results of each dataset. For comparison, we show in the top half of the table, QRec's in-domain performance on the target. We argue that QRec achieves remarkable generalizability across datasets with on average only 0.5% reduction in AUC, compared to QRec trained using the seen user-seen item subset in the target domain.

the-art neural architectures for recommendation. For the regular in-domain recommendation, across five datasets, QRec generally falls short against the best of the selected baselines and outperforms purely popularity-based baselines by up to 6%. What is quite amazing is that QRec can generally explain 85% to 93% of the AUC score of the best-performing baseline (and in case of Amazon-Cloth dataset, is the best performer; beats well-established NCF and BPR for both Amazon-Cloth and Amazon Sport), while only taking a fraction of the training time (less by a factor of 3×–10× see table 5 in the appendix). We discuss zero-shot cases next.

*In-domain, zero-shot recommendation (RQ2):* For the zero-shot in-domain recommendation setting, we train QRec on the seen user-seen item subset of the domain. We make predictions using eq. (7), on the unseen user-unseen item of the same domain. The first interesting observation (see table 2): the performance of QRec in the zero-shot case matches the performance of QRec in the regular in-domain case. This finding implies that the learned representations of the activity percentiles have significant predictive power, on unseen data within the same domain. While QRec can still outperform the popularity-based baseline by up to 5%, with QRec's performance gain over MostPop is higher for the most active users as seen in the appendix, fig. 2. Still, the performance of the popularity-based baseline is impressive. We suspect the reason is that the popularity bias (Sankar et al., 2021; Zhang et al., 2021) is prevalent in the original datasets and thus users are much more likely to interact with popular items regardless of their general preference.

*Cross-domain, zero-shot recommendation (RQ2):* For the zero-shot cross-domain recommendation setting, we train QRec on the seen user-seen item training subset of a source domain $M$. We make predictions using eq. (7), on a different, target domain $M'$ (see table 3). To ensure that we have an apples-to-apples comparision, we do not use the entire target dataset for computing the results. Instead, for the target, we use the same subset that we used for training the baseline in-domain recommendation case. Thus, we can ask the question: *what might have been improvements in the results had we chosen to re-train a state-of-the art neural recommendation system on the target domain?* A further factor that is QRec requires no re-training, while the baselines can take significant training time per epoch (see table 5 in the appendix). QRec shows excellent generalization performance across the board, with exceptions when transferring from MovieLens to other datasets, while two Amazon datasets achieve the best pairwise cross-domain zero-shot recommendation result. The likely reason is the difference in dataset statistics, as shown in table 4. Therefore, their marginal activity distribution may exhibit different characteristics. These experimental results empirically show that QRec is highly transferrable, not only within domains but also across datasets.

*Effect of Training data (RQ3):* In this section, we examine (see fig. 1) how the amount of training data influences both the in-domain recommendation and the in-domain zero-shot generalization performance. We vary the percentage of interaction used as training for each user. Here, 100%, refers to the dataset that was used for the standard, in-domain training. Numbers less than 100%, say 20%

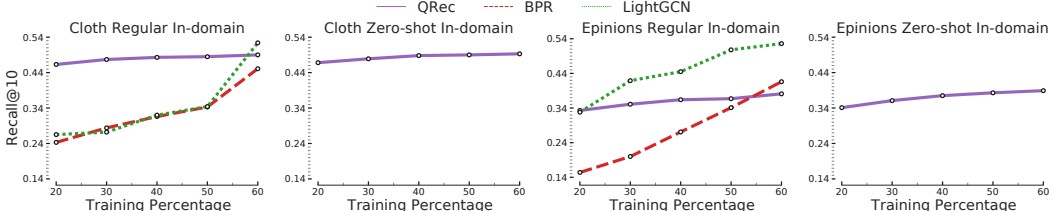

Figure 1: In-domain and zero-shot in-domain recommendation result vs training percentage of each user's interaction history. We use the same test set as the regular in-domain recommendation (RQ1) and zero-shot in-domain recommendation (RQ2). For each user, we vary the percentage of interactions for calculating percentiles and training. When the training percentage is 60, it corresponds to experimental results in Table 2. We keep the minimal training percentage to 20% due to the 5-core setting we adopt during preprocessing. QRec shows stable performance across all training percentages, while the performance of the selected baselines fluctuates significantly. QRec outperforms both state-of-the-art baselines on the 20% training percentage.

refers to sampling 20% of the original dataset using stratified sampling to ensure that the marginal distributions are similar to the original.

The key observation in fig. 1 is that the regular in-domain and zero-shot in-domain performance of QRec on both datasets is stable compared to state-of-the-art neural recommender models, whose performance drops significantly with decrease in training set size. The implication is that QRec can work well as the recommender system for smaller unseen datasets, the ideal use case for pre-trained recommender models. Note that when the training percentage drops to 20%, QRec outperforms the selected stat-of-the-art models by 74% and 1% on Amazon-cloth and Epinions respectively. We attribute the difference in performance to the fact that Epinions is denser than Amazon-Cloth.

## 5.4 DISCUSSION

QRec shows excellent potential as a pre-trained recommendation model, capturing on average around 90% of the state-of-the-art neural recommenders' performance with a fraction of the training time and model size while capable of in-domain and cross-domain zero-shot transfer to other datasets using only interaction data. This suggests the existence of cross-domain, statistical patterns of user-item interactions; *i.e.,* two users *across domains* are similar if they share similar statistical interaction patterns with items, regardless of the user and items' descriptions.Thus, QRec can both serve as an initial recommendation model when the data in the domain is sparse and also as a basis for developing more complex neural recommendation models when more data are available.

QRec examines the marginal distributions and learns to represent users and items based on their activity percentiles, with excellent cross and in-domain zero-shot prediction results. However, the marginal distributions may not be sufficient to capture the complex user-item interaction patterns. In addition to the marginal, identifying structure in the joint distribution of user-item interactions, and learning transferable representations of this structure could potentially further boost the performance of pre-trained recommendation models.

## 6 CONCLUSION

Inspired by the impact of pre-trained language/vision models, we explored the possibility of pre-trained recommender models that can support building recommender systems in new domains, with minimal or no retraining, without the use of any auxiliary user or item information. Our insight was to identify the importance of the ubiquitous heavy-tailed marginal distribution of user/item activities across recommendation datasets. We proposed QRec, to learn the representations from the marginal activity quantiles of the users and items. We represent users (items) based on the statistical representations of items (users) they interact with. We used positional encoding to impose order on the learned representations of percentiles and devised a regularization loss to further smooth the statistical embeddings. Through extensive experiments, we show that QRec can not only retain up to 93% of the state-of-the-art neural recommender performance with a fraction of the training time and model size but also achieve excellent generalization performance on unseen users and items (*i.e.,* zero-shot setting) within and across different domains.

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

| DATASET | EPINIONS | YELP | MOVIELENS | CLOTH | SPORT |
|---|---|---|---|---|---|
| # Seen Users | 5,485 | 9,481 | 3,013 | 113,233 | 20,865 |
| # Seen Items | 5,469 | 5,644 | 1,557 | 45,353 | 10,609 |
| # S-S Interactions | 73,459 | 133, 227 | 211,059 | 2,201,273 | 153,223 |
| # Unseen Users | 5,699 | 9,585 | 3,004 | 112,057 | 22,343 |
| # Unseen Items | 5,613 | 5,549 | 1,571 | 44,583 | 11,292 |
| # U-U Interactions | 77,838 | 136,557 | 206,096 | 2,194,134 | 165,836 |
| Density | 0.0024 | 0.0025 | 0.0443 | 0.0009 | 0.0007 |

Table 4: Dataset statistics

# A APPENDIX

## A.1 POSITIONAL ENCODING

We follow the same fixed positional encoding scheme as in Bahdanau et al. (2014) to impose order on the learned representations of percentiles. Formally, we define the positional encoding as:

$$\boldsymbol{T}_{r,2i} = \sin\left(\frac{r}{n^{2i/d}}\right), \quad \boldsymbol{T}_{r,2i+1} = \cos\left(\frac{r}{n^{2i/d}}\right) \tag{8}$$

where $r$ is the percentile of the user/item activity, $n$ is a pre-defined scalar, $d$ is the dimension of the percentile representation, and $i$ is the dimension index. We use $n = 100$ in our experiments.

## A.2 DATASET DESCRIPTION

EPINIONS : product ratings from Epinions; we binarize the explicit ratings by keeping ratings of 3 or higher.

YELP : user ratings on local businesses located in the state of Arizona, obtained from Yelp dataset challenge[4].

MOVIELENS 1M : widely used movie rating dataset from MovieLens[5]. The dataset contains one million ratings where each user has at least 20 ratings;

AMAZON-SPORT, AMAZON-CLOTH : sport and cloth categories of the amazon product review dataset collected by Ni et al. (2019);

As a preprocessing step, we also binarize the explicit ratings on Yelp, MovieLens, and the two Amazon review datasets, by keeping ratings of 3 or higher. Detailed dataset statistics are in table 4. The density is computed as the ratio of the number of interactions to the number of users times the number of items.

## A.3 BASELINES DESCRIPTION

In this section, we describe the baselines used in our experiments. For the state-of-the-art neural recommenders, we use the implementations provided by the authors and tune the hyper-parameters on the validation set. We report the best results on the test set.

BPR RENDLE ET AL. (2009): neural CF model with matrix factorization and Bayesian Personalized Ranking loss.

NCF HE ET AL. (2017): neural CF model with non-linear neural layers between the user and item latent embeddings.

NGCF WANG ET AL. (2019B): graph-based NCF model with embedding propagation layers on the user-item interaction graph.

LIGHTGCN HE ET AL. (2020): state-of-the-art graph neural network models for recommendation with simplified GNN designs.

---

[4]https://www.yelp.com/dataset
[5]https://grouplens.org/datasets/movielens/1m/

| DATASET | EPINIONS | YELP | MOVIELENS | CLOTH | SPORT |
|---|---|---|---|---|---|
| # of Interactions | 73,459 | 133, 227 | 211,059 | 2,201,273 | 153,223 |
| BPR | 5,008.9 | 9,047.9 | 5,516.1 | 264,195.8 | 16,156.9 |
| NCF | 5,657.4 | 9,762.9 | 6,013.3 | 287,469.6 | 17,560.2 |
| NGCF | 5,173.9 | 9,466.1 | 5,967.2 | 280,751.9 | 17,194.4 |
| LightGCN | 5,116.6 | 9,122.7 | 5,901.2 | 280,207.1 | 17,231.8 |
| QRec | 2,966.5 | 4,553.7 | 3,987.3 | 19,464.1 | 5,106.8 |

Table 5: Training time per epoch in milliseconds of different models on different datasets. QRec is able to reduce the training time by 28% to 93% while achieving competitive performance.

MOSTPOP RENDLE ET AL. (2009); HE ET AL. (2017); KANAGAWA ET AL. (2019): create rank lists by ranking items based on their interaction counts. We only report the results of Recall@10 and NDCG@10, since it does not produce a prediction score for a given user-item pair.

## A.4 EXPERIMENTAL DETAILS

Here we provide additional details of our experiments, following § 5.2. We use the same experimental setup for all models, *e.g.,* batch size and embedding size. We set the batch size to 1024 and the embedding size to 64. We use Adam optimizer and tune the learning rate in the range $\{10^{-4}, 10^{-3}, 10^{-2}\}$ and set the weight decay to $10^{-5}$. We use dropout regularization with a rate of 0.3. We tune the balance hyper-parameter $\lambda$ in the range $\{10^{-1}, 10^{0}, 10^{1}\}$.

## A.5 COMPLEXITY ANALYSIS

We compare the time cost training QRec per epoch with other state-of-the-art neural recommenders in table 5. We use the same experimental settings on all models, *e.g.,* batch size and embedding size. We observe that QRec is significantly faster than other state-of-the-art neural recommenders, reducing the training time by up to 93% while achieving comparable performance. In addition, the training time of QRec does not scale with the number of user, items, and interactions, making it suitable for large-scale recommendation systems.

We also compare the model size (*i.e.,* number of learnable parameters) of QRec with other state-of-the-art neural recommenders. The trainable parameters of QRec only consist of the learnable look-up matrix $Q$, learnable weight matrices $W_u$ and $W_v$, and the learnable weight matrix for the attention function $h$. The model size of QRec is essentially $O(d^2)$, whereas the state-of-the-art neural recommenders have a model size of $O(d|\mathbb{V}| + d|\mathbb{U}| + d^2)$ since they have trainable parameters for each user and item. Almost in all cases, $|\mathbb{V}|$ and $|\mathbb{U}|$ are significantly larger than $d$. Therefore, QRec is able to achieve competitive performance against state-of-the-art neural recommenders with a significantly smaller model size.

## A.6 CROSS-DOMAIN ZERO-SHOT RECOMMENDATION RESULTS ON UNSEEN USERS AND UNSEEN ITEMS

In this complementary experiment, we show the results of cross-domain zero-shot recommendation on the test set of unseen users-unseen items in table 6. Note that the test set of this experiment corresponds to the test set of in-domain zero-shot recommendation setting, unlike results in table 3, where the test set corresponds to the regular in-domain recommendation setting. We see that QRec also exhibits great generalizability on the unseen users-unseen items test set, with up to 1.5% and on average 1% performance deduction compared to the in-domain zero-shot results. This experiment further demonstrates that QRec is able to learn transferable representations that can be used to generalize to all users and items in a different domain.

## A.7 ABLATION STUDY

In this experiment, we study the effectiveness of the positional encoding (*i.e.,* enforcing order on the percentile representations) and the regularization loss (*i.e.,* smoothness of the percentile repre-

| S → T | EPINIONS | | | YELP | | | MOVIELENS | | | CLOTH | | | SPORT | | |
|---|---|---|---|---|---|---|---|---|---|---|---|---|---|---|---|
| Metric | AUC | R@10 | N@10 | AUC | R@10 | N@10 | AUC | R@10 | N@10 | AUC | R@10 | N@10 | AUC | R@10 | N@10 |
| REFERENCE: IN-DOMAIN ZERO-SHOT RECOMMENDATION | | | | | | | | | | | | | | | |
| MostPop | —— | 0.371 | 0.217 | —— | 0.420 | 0.238 | —— | 0.510 | 0.294 | —— | 0.487 | 0.332 | —— | 0.370 | 0.213 |
| QRec | 0.712 | 0.389 | 0.220 | 0.753 | 0.428 | 0.242 | 0.833 | 0.513 | 0.296 | 0.772 | 0.493 | 0.336 | 0.694 | 0.379 | 0.214 |
| CROSS-DOMAIN RECOMMENDATION, SOURCE → TARGET | | | | | | | | | | | | | | | |
| Epinions*→ | —— | —— | —— | 0.751 | 0.425 | 0.239 | 0.813 | 0.449 | 0.263 | 0.697 | 0.428 | 0.275 | 0.692 | 0.376 | 0.217 |
| Yelp*→ | **0.700** | **0.382** | **0.220** | —— | —— | —— | 0.807 | 0.490 | 0.286 | 0.756 | 0.488 | 0.332 | 0.692 | 0.375 | 0.217 |
| MovieLens*→ | 0.664 | 0.360 | 0.202 | 0.727 | 0.405 | 0.226 | —— | —— | —— | 0.726 | 0.484 | 0.331 | 0.684 | 0.370 | 0.208 |
| Cloth*→ | 0.697 | 0.378 | 0.219 | **0.754** | 0.423 | **0.239** | 0.825 | 0.500 | 0.282 | —— | —— | —— | **0.698** | **0.378** | **0.218** |
| Sport*→ | 0.699 | 0.381 | 0.221 | 0.752 | **0.424** | 0.239 | **0.826** | **0.503** | **0.288** | **0.769** | **0.491** | **0.335** | —— | —— | —— |

Table 6: *Cross-domain, zero-shot:* Recommendation results of cross-domain zero-shot transfer on unseen users and unseen items. $S \rightarrow T$ means we pre-train the recommender system using S's data(columns) and directly test on T (rows). Similar to table 3, we train the model using the seen users-seen items training set of the source domain. However, different from table 3, where the zero-shot transfer test set is the seen users-seen items test set, the test set here is the unseen users-unseen items test set. Therefore, the results in this table correspond to the in-domain zero-shot results in table 2. The purpose of this experiment is to give a more complete picture of the cross-domain zero-shot transfer results and provide more evidence that QRec is highly generalizable to unseen users and unseen items. QRec is able to achieve 1% performance reduction on average compared to the in-domain zero-shot results.

| Scenario | REGULAR IN-DOMAIN RECOMMENDATION | | | | | | IN-DOMAIN ZERO-SHOT RECOMMENDATION | | | | | |
|---|---|---|---|---|---|---|---|---|---|---|---|---|
| Dataset | EPINIONS | | | CLOTH | | | EPINIONS | | | CLOTH | | |
| Metric | AUC | R@10 | N@10 | AUC | R@10 | N@10 | AUC | R@10 | N@10 | AUC | R@10 | N@10 |
| QRec | 0.706 | 0.381 | 0.214 | 0.772 | 0.490 | 0.332 | 0.712 | 0.389 | 0.220 | 0.772 | 0.493 | 0.336 |
| w\o reg-loss | 0.706 | 0.381 | 0.213 | 0.768 | 0.489 | 0.325 | 0.706 | 0.389 | 0.220 | 0.770 | 0.493 | 0.329 |
| w\o positional | 0.706 | 0.382 | 0.217 | 0.755 | 0.490 | 0.330 | 0.704 | 0.387 | 0.221 | 0.753 | 0.493 | 0.335 |
| w\o reg + pos | 0.700 | 0.381 | 0.216 | 0.757 | 0.485 | 0.329 | 0.704 | 0.383 | 0.217 | 0.760 | 0.489 | 0.334 |

Table 7: Ablation study of QRec . We report the results of QRec with different components removed. w\o reg-loss means without the regularization loss, w\o positional means without the positional encoding, and w\o reg + pos means without both the weighting regularization loss and the positional encoding. We see that both the regularization loss and the positional encoding improve the performance of QRec, especially for the in-domain zero-shot recommendation.

sentations) in QRec. We report the results in table 7. We see that both the regularization loss and the positional encoding improve the performance of QRec. In particular, the regularization loss, while having no substantial effect on the regular in-domain recommendation results, improves the performance of QRec by up to 1% in AUC and 2% in NDCG for in-domain zero-shot recommendation. This shows that the regularization loss effectively increases the generalizability of QRec by smoothing the percentile representations. The positional encoding, on the other hand, improves the performance of QRec by up to 2% in AUC for regular in-domain recommendation and 1% in AUC for in-domain zero-shot recommendation results. This indicates that imposing order on the percentile representations is beneficial to the overall model performance. Together, they improve the model performance by 2% in AUC for regular in-domain recommendation and 2% in AUC for in-domain zero-shot recommendation.

## A.8   RECOMMENDATION RESULTS OF DIFFERENT USER GROUPS

Both QRec and the MostPop baseline rely on the marginal distribution of user/item activities. This experiment is to better understand the benefits of learning refined statistical representations. From fig. 2, we can see that when the number of interactions of the users increases, QRec shows greater performance gain, apart from the 5th group results on sport zero-shot in-domain recommendation. The potential reason for this could be the difference in dataset statistics between the sport regular in-domain and zero-shot in-domain dataset. On the Epinions dataset, QRec beats the MostPop by a large margin on the most interactive users, while having similar performance on the least interactive users. The possible reason is that when a user has few interactions, the constructed representations based on statistics might not be noisy and thus are less indicative of the users' preference. For users

Figure 2: In-domain and zero-shot in-domain recommendation results of different user groups. We divide the users into 5 equal-size groups based on their number of interactions and evaluate the recommendation performance within each groups. User group 1 consist of the least interactive users, while user group 5 consists of the most interactive users. We see that QRec has a larger margin of improvement over MostPop for more interactive users.

with higher interaction counts, where QRec has larger performance gains, the user representations learned by QRec can reflect more of the users' preferences. This experiment shows that for users with more interactions, QRec is better at capturing the users' preferences.

