# OpenReview forum: "Pre-trained Neural Recommenders: Learning Statistical Representations for Zero-shot Recommender Systems"
_ICLR.cc/2024/Conference — ICLR 2024 Conference Withdrawn Submission_

### Official Review · Reviewer_aga8 · 2023-10-30

**Soundness:** 2 fair
**Presentation:** 3 good
**Contribution:** 2 fair
**Rating:** 3
**Confidence:** 4

**Summary:**

This paper proposes QRec, a zero-shot recommendation model which learns cross-domain user/item representations by leveraging the activity quantiles. The authors demonstrate that, without using any auxiliary user/item information, such framework can generalize very well to unseen users unseen items within the dataset and across different datasets. Extensive experiments on a number of large-scale, real-world datasets show that QRec is able to outperform the MostPop baseline.

**Strengths:**

- Representing users and items using the user/item activity quantiles is an interesting idea and is, to the best of my knowledge, novel.
- The paper is well-written with a clear structure and very easy to follow.
- The experiments are quite extensive, and the ablation study is also well designed.
- QRec is able to outperform MostPop, which validates that the quantile-based information is useful.

**Weaknesses:**

- QRec is only marginally better than MostPop.
- From my understanding, since QRec is only marginally better MostPop, it may indicate that the model is basically learning to relate items at specific popularity quantiles to users at particular quantiles.
- QRec is not compared against other zero-shot/few-shot recommendation models.
- The authors fail to clearly explain why quantile-based representations is generalizable across recommendation domains.

**Questions:**

- Is there any way that we can visualize the statistical patterns that QRec has learned? For example, maybe we can investigate what patterns the model can discover when recommending the least popular item to the most frequent user.
- How personalized are the recommendations if a number of users/items will have the same representations due to being in the same activity quantiles?

---

### Official Review · Reviewer_YAJj · 2023-11-01

**Soundness:** 2 fair
**Presentation:** 4 excellent
**Contribution:** 1 poor
**Rating:** 3
**Confidence:** 4

**Summary:**

The paper proposes a zero shot approach to collaborative filtering. A generic percentile representation is learned on a given dataset. User and item embeddings are then derived using this percentile representation using counts of interactions in each percentile. This makes the approach applicable to new datasets without re-training.

**Strengths:**

The paper is well written and I found the percentile embedding approach interesting and quite innovative.  Experiments are conducted on multiple real world datasets and evaluate zero shot in-domain and cross-domain generalisation.

**Weaknesses:**

While the idea of a foundational recommender model that facilitates zero shot generalisation is definitely exciting and ambitious, I don't see how the proposed approach can capture sufficient information for personalization. Considering equations 1 and 2 this approach will be able to capture different activity patterns for users and items, such as popular vs unpopular items but it will not capture which of the popular/unpopular items a given user prefers. This is further validated by tables 2 and 3 where the performance of QRec is essentially identical to MostPop. So in essence I think the percentile embeddings are just capturing fine grained popularity statistics. Given these results I think this approach is of limited utility and needs further development.

**Questions:**

I think a more thorough comparison with MostPop is needed given the similarity in results. Have you looked at the top recommended item overlap between QRec and MostPop? I suspect it will be quite high. Diversity of recommended item lists across users would also be good to analyze, is QRec mostly recommending the same items for each user?

---

### Official Review · Reviewer_bhvm · 2023-11-01

**Soundness:** 2 fair
**Presentation:** 3 good
**Contribution:** 2 fair
**Rating:** 3
**Confidence:** 4

**Summary:**

The paper presents a recommendation algorithm by learning user and item representations from the distribution of users' interaction on the item popularity quantiles. The assumption is that users preference patterns in terms of the popularity bin are the same across different domains. Each percentile bin is learned as an embedding vector, and then each user and item representation is learned as an attentive aggregation of their corresponding percentile bins based on their interactions. Experiments are conducted on several real-world datasets to test the in-domain and cross-domain performance of the proposed method.

**Strengths:**

+ Learning user and item embeddings from the quantile distribution is an interesting idea to explore.
+ The proposed method are tested on five real-world datasets.
+ The experiments tested both in-domain and cross-domain performance.

**Weaknesses:**

- The proposed method relies on a key but hidden assumption: Different users may have different preference patterns over the popularity-degree of items, e.g., if we split the items into 100 bins according to their popularity, some users may prefer items in bin #2, 16 and 56, while another users may prefer items in bin #23, 48, 96, etc. Then the research makes a key assumption that users preference patterns in terms of the popularity bin are the same across different domains. However, this assumption may not be true, for example, it is quite possible that a user likes popular items on the movie domain but likes niche items on the music or product domain. To set up a persuading motivation for the proposed method, it is important to at least provide case studies using real-world datasets to test if or not this assumption holds, under what condition does it hold, and by what degree it holds.

- Problem definition 1 and 2 claims "without training the scoring function on M' or M*". However, the proposed method is still constructing a scoring function based on M' and M*, using hand-craft statistical functions instead of learned function, but no matter the function is hand-crafted or learned, it needs to use the information in M' and M* for prediction, and thus there really is no fundamental difference, the only difference is wether the function is hand-crafted or machine-learned. It would be better if the paper further clarifies the problem definition.

- Baseline methods are too old and weak, and did not include any of the other pre-trained models for recommendation, such as pre-trained language model based recommendation methods. Given that the proposed method aims at pre-trained recommendation models, it's important and necessary to compare with other existing pre-trained recommendation methods such as pre-trained language models for recommendation.

- Many of the conclusions derived in the experimentation section is based on the ACU metric. However, AUC is not a realistic evaluation metric for recommendation tasks. It would be better to derive the conclusions based on other more realistic metrics such as Hit Ratio and NDCG.

**Questions:**

After pre-processing each dataset, what does the experiment do with the seen user - unseen item group and the unseen user - seen item group? Are these data points simply dropped?

Table 3 only includes the MostPop baseline, it would be better to include other baselines, and also, as mentioned above, include more recent methods especially pre-trained models for recommendation.

---

### Official Review · Reviewer_b2di · 2023-11-06

**Soundness:** 2 fair
**Presentation:** 3 good
**Contribution:** 2 fair
**Rating:** 5
**Confidence:** 3

**Summary:**

Authors propose a zero-shot neural recommendation method which relies on the activity percentile embedding of the user and item as opposed to the categorical user and item embedding, commonly used in neural recommender systems. Authors claim that using the activity percentile embedding generalizes to unseen users and items on different tasks (cross-domain and zero-short recommendations). Given the activity embedding for user and item, the method is trained via the standard BPR pairwise loss. Authors conduct experiments on multiple public recsys benchmark datasets.

**Strengths:**

- The presented method is simple and easy to implement, which is an important factor for actual production systems.
- The proposed method relies only on the binary interaction matrix, and does not require any additional metadata, which can be useful when user and item metadata is not available.
- The training time for the proposed method is significantly lower than the existing baseline methods, which is a great practical advantage.

**Weaknesses:**

- In most of the practical setting, user and item metadata information is available, I am not sure how can practitioners adapt the existing method to use metadata, given it only relies on the interaction data.
- For the zero-short setup, only popularity-based baseline is used, I would expect some model based baseline, for a fair comparison.

**Questions:**

- For the zero-shot setting, when there is no user/item interaction is available, how would you compute the activity percentile for that user/item?
- How can you adapt the existing method to utilize the user/item metadata?